# Scorpionism in Brazil: Space-time approach and risk areas in 2012 to 2024

Alec Brian Lacerda[1]*, Denise Cândido[2], Thiago Salomão de Azevedo[3,4], Fan Hui Wen[2], Gisele Dias Freitas[5], Flávio Santos Dourado[6], Francisco Chiaravalloti Neto[1]

1 School of Public Health, University of São Paulo, São Paulo, Brazil, 2 Butantan Institute, São Paulo, Brazil, 3 Secretary of Health, Municipality of Santa Barbara d'Oeste, São Paulo, Brazil, 4 Department of Biodiversity, Institute of Biosciences, UNESP, Rio Claro, São Paulo, Brazil, 5 State Health Department - Epidemiological Surveillance Center (CVE), São Paulo, Brazil, 6 Brazilian Ministry of Health, Brasília, Distrito Federal, Brazil

* alec.lacerda@usp.br

## Abstract

### Background

Scorpionism is a neglected public health problem in Brazil and there is currently a significant increase in the number of cases. In Brazil, the Tityus genus is the biggest cause of accidents and increases the risk of death in children under the age of 10. The aim of this study is to identify high and low risk areas for scorpion accidents in Brazil.

### Methodology

This is an ecological and descriptive study of the occurrence of scorpionism in Brazil and in high-risk municipalities between 2012 and 2024. The analysis included all 5,570 municipalities of Brazil. Bayesian incidence rates were calculated and standardized by age group and sex. Spatial, space-time, temporal, seasonal and time trend scanning techniques were used to identify high and low risk clusters in Brazilian municipalities. The Gini coefficient function was selected to remove hierarchical clustering detection and to identify the best population percentage (%) for the data sample. Socio-demographic, environmental and climatic variables were chosen and compared between the municipalities within the high and low risk clusters to assess the indicators in different realities.

### Principal Findings

A total of 1,729,023 cases of scorpionism and 1,230 deaths were reported across Brazil's 5,570 municipalities between 2012 and 2024. The incidence rate rose from of 31.8 per 100,000 inhabitants in 2012 to 142.82 per 100,000 inhabitants in 2024 (349% increase). The data shows that more children die and the older adults suffer

**Data availability statement:** The data is available in aggregate form, without any personal or identifiable information, for the entire period and at all study sites at https://zenodo.org/records/17298805.

**Funding:** This study was financed in part by the Coordenação de Aperfeiçoamento de Pessoal de Nível Superior - Brasil (CAPES) - Finance Code 001. This work was also supported by the National Council for Scientific and Technological Development (CNPQ) (research productivity grant - level 1C, number 304391/2022-0 to FCN). The funders had no role in the study design, data collection and analysis, decision to publish, or preparation of the manuscript.

**Competing interests:** The authors have declared that no competing interests exist.

the most from accidents. The regions of Minas Gerais, São Paulo and Bahia were the areas most affected by scorpionism, with high-risk clusters and an upward trend over time. The northern region showed the opposite pattern.

## Conclusions

More studies are needed to understand why these accidents happen in these regions, in order to support policies, surveillance actions and the control and monitoring of this health problem.

### Author summary

Scorpion stings are becoming a growing health problem in Brazil, but the issue often gets overlooked. This study looked at where the risk of scorpion stings is highest and lowest across the country. Between 2012 and 2024, researchers examined data from all 5,570 cities in Brazil using mapping and statistics to spot risk areas. They found that scorpion sting cases rose by 349%, with over 1.7 million cases and 1,230 deaths during that time. Children under 10 were the most at risk, but many older adults were affected too. The states hit hardest were Minas Gerais, São Paulo, and Bahia, where cases have continued to rise. Meanwhile, the northern region showed different trends. The study highlights the urgent need to better understand why these stings are increasing and calls for more research to help guide public health actions and improve prevention and monitoring efforts.

## Introduction

Scorpion envenoming or scorpionism is an important public health problem, especially in tropical countries. It affects more than one million people and leads to approximately 3,000 deaths yearly [1]. In the world, the main countries endemic with scorpionism are Mexico, Brazil, Algeria, Iran, Tunisia, Morocco, Saudi Arabia and Turkey [2], with Brazil being the second largest in number of cases and the largest in number of deaths. In Brazil, the incidence of scorpionism has been increasing significantly, likely due to environmental factors that promote scorpion proliferation in urban areas and mandatory notification in national information systems since 1997 [3]. Over 1,250,000 cases were reported between 2007 and 2020 in Brazil, with significant incidences in the northeast and southeast regions of the country, highlighting the need for continued monitoring [4].

The species that most frequently cause scorpion sting envenoming in Brazil belong to the genus *Tityus* (Family Buthidae): *T. serrulatus* (yellow scorpion), *T. bahiensis* (brown or black scorpion), *T. stigmurus* (yellow scorpion of the northeast), and *T. obscurus* - formerly known as *T. paraensis* - (black scorpion of the Amazon) [5]. Of these, *T. serrulatus,* found in at least 70% of Brazilian states [6], is responsible for the majority of accidents and the most severe envenoming, mainly in children [4,7].

Venom from scorpion stings can cause a wide range of envenoming spectra, with 90% of cases considered mild and good prognoses [8]. The venom's amount and composition and the victim's clinical condition directly influence the outcome, with children aged < 9 years being more severely affected [9].

Recent studies in Brazil underscore the importance of epidemiological surveillance, especially in regions where scorpionism is more frequent and severe. Spatial analyses have identified risk areas and specific management strategies for the most affected localities [10–13]. Analysis of spatial clusters favors the identification of priority regions for epidemiological and entomological control and monitoring, mainly for the correct allocation of resources and the effectiveness of health actions. [14] and [15] identified high-risk areas in São Paulo and associated them with climatic and environmental conditions that may explain the increased scorpion accident rates in specific regions.

Climate change is an important factor in the migration of species, adaptation to new habitats and contact with human beings, and these are fundamental characteristics to work on when monitoring these accidents. Some recent studies have presented important results for new environmental and climate management policies to be incorporated into scorpion accident monitoring practices [15–17].

This study aimed to identify clusters of high- and low-risk scorpionic accidents in Brazil and associate them with demographic, climatic, environmental, and geomorphological factors. The results may enhance understanding of this important and neglected public health issue, support local surveillance in identifying accident-prone areas, and contribute to the development of strategies for synanthropic control.

## Methodology

This is an ecological and descriptive study of the occurrence of scorpionism in Brazil and in high-risk municipalities between 2012 and 2024. The analysis included all 5,570 municipalities of Brazil (Fig 1).

Data on scorpion sting injuries between 2012 and 2024 were obtained from the Notifiable Diseases Information System (SINAN), while population data was obtained from the Brazilian Institute of Geography and Statistics (IBGE). The map files (shapefiles) can be accessed freely and free of charge through the IBGE territorial grids [18]. Duplicate reports were analyzed using Reclink3 [19] with a probabilistic method to avoid overestimating the analyses.

Descriptive analysis was performed on scorpion sting cases and the population, characterized according to sex and age group in time and space. Overall incidence rates of scorpionism were calculated by sex and age. Direct standardization was performed according to age group and sex, using the average population for the period as the standard population.

Annual scorpionism incidence rates were calculated for the municipalities in Brazil and presented as Local Empirical Bayesian rates with contiguity neighborhood criterion type I (Queen). These rates were age standardized. This technique considers the average rate of the neighborhoods in each area, enabling more stable estimates for regions with few inhabitants. This technic models the random fluctuation in areas with low population numbers by adjusting incidence rates based on the characteristics of nearby or similar areas. The results were presented in choropleth maps, including information on municipalities with scorpionic accidents. Maps were generated using *R* [20]. The main packages used were *tidyverse*, *cowplot* and *ggspatial*.

Scanning techniques were used to detect spatial, temporal, seasonal, and spatiotemporal clusters and the spatial variation of the temporal trend, considering the discrete retrospective Poisson model. As described by [21], spatial scan establishes a circular window around an area's centroid, varying the radius size and alternating between zero and its geographically pre-established limit (default 50%). The same occurs with the spatiotemporal scan, with observation of time in a cylindrical window based on the period. The clusters of the spatial variation of the temporal tendency used a spatial window with the temporal tendency measured inside and outside this window, without using the relative risk (RR) as the main factor.

PLOS Neglected Tropical Diseases

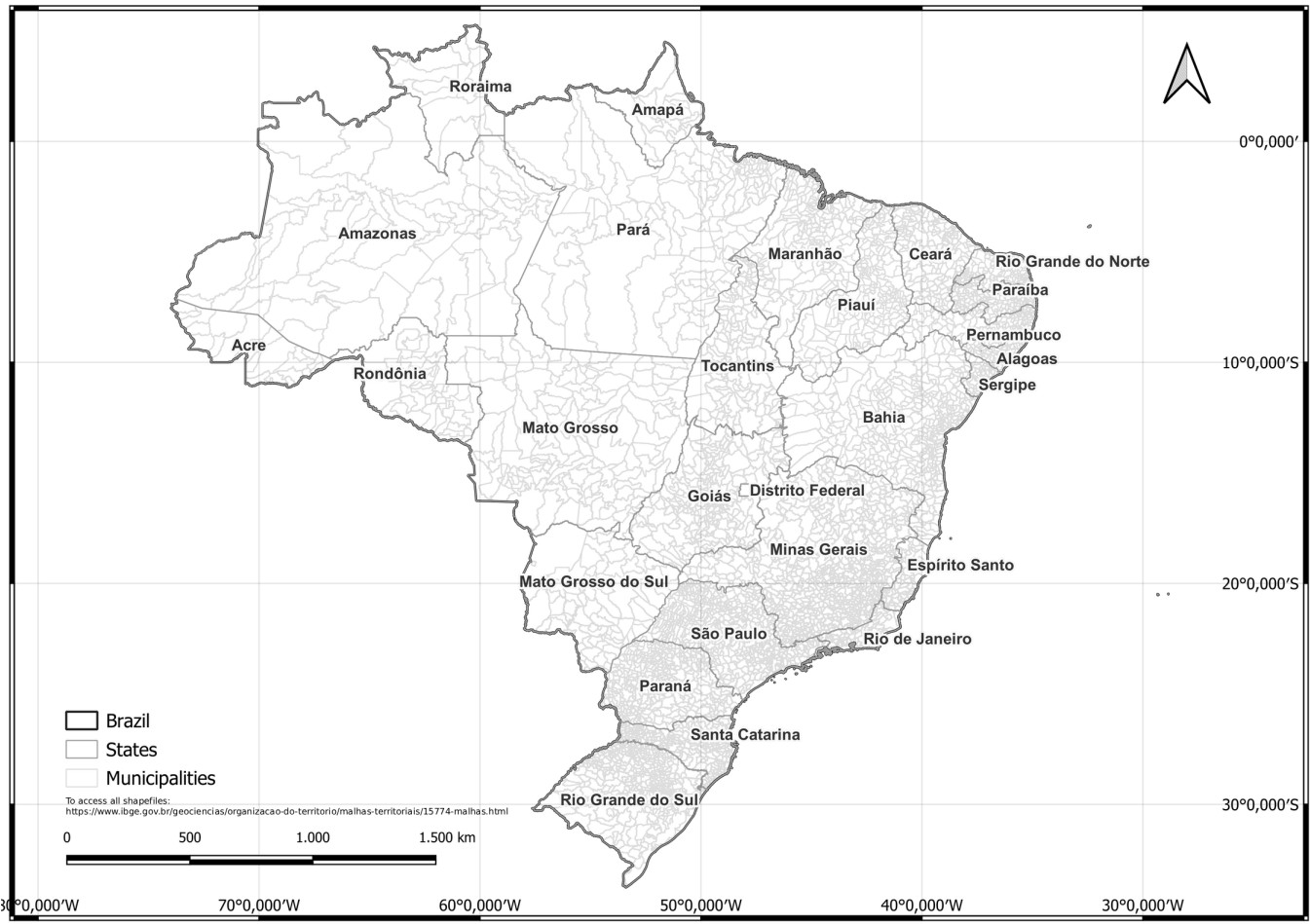

**Fig 1. Map of Brazil.** Source of base layer maps (states and municipality boundary): https://www.ibge.gov.br/geociencias/organizacao-do-territorio/malhas-territoriais/15774-malhas.html.

To optimize the identification of significant spatial clusters while avoiding hierarchical bias, we applied the Gini coefficient function within SaTScan, following the methodology proposed by [22]. This approach allowed for the determination of an appropriate maximum population size of 8% for the spatial scanning window. For the temporal dimension, the scanning window was set to a maximum of 50% of the study period. Cluster significance was assessed using a Monte Carlo simulation with 999 replications, and a 5% significance threshold was adopted [Kulldorff, 2021]. Finally, the results highlighted the RR, ideal population, cases within the clusters, and p-values.

For statistical analysis of the comparison of means, a t-test of two means was used, a hypothesis test that can be used to compare two means with the null hypothesis that the means are equal. The type of variance test was identified using the F test to categorize the t-test for different or equivalent variances. The database comprised variables for municipalities composing high- and low-risk clusters. The test compares the means of a specific variable for these two groups, showing, first, whether there is a statistically significant difference between these means and, second, whether the variable represents the risk or protection against the occurrence of scorpion accidents.

The variables chosen were: Minimum temperature (%), Maximum temperature (°C), Average precipitation (mm), Temperature range, Sewage treatment (%), Water treatment (%), Garbage collection, Alphabetization, Urbanization

(%), Natural vegetation (%), and Normalized Difference Vegetation Index (NDVI). These variables, proven to be crucial for investigating scorpion accidents in recent studies [13,23], can contribute to the analysis as explanatory tools for the events. The variables were obtained from freely accessible information systems. The environmental and climatic variables were taken from MapBiomas and WorldClim and the socio-demographic data was collected from IBGE systems.

## Results

A total of 1,729,023 cases of scorpionism and 1,230 deaths were reported across Brazil's 5,570 municipalities between 2012 and 2024. The incidence rate rose from of 31.8 per 100,000 inhabitants in 2012 to 142.82 per 100,000 inhabitants in 2024 (349% increase). The state of Alagoas was the most incident, with more than 270 cases per 100,000 inhabitant-years over the 13-year study period. Table 1 presents this information stratified by sex and age group, and Table 2 stratified by sex and region of Brazil.

About Table 1, between 2012 and 2024, scorpion sting cases in Brazil increased with age, with higher rates among males as age advanced. Females had similar rates to males; however, they presented higher incidence in the 0–19 age group and less growth intensity in the advancing age group. Among those aged ≥ 60 years, incidence rates were higher in both sex, representing the increase in incidence with age. The older adult age group represented 16.6% of the cases of scorpionism in Brazil, which was 6.2% higher compared to children aged 0–9 years (10.4%). Incidence rates among older adults were 63% higher than that in children aged 0–9 years, highlighting an important situation for the epidemiological analysis of risk populations.

Mortality rates were higher among males across all age groups. In both sexes, the lowest mortality impact was observed in adults aged 10–39, while the highest mortality occurred in children aged 0–9, followed by older adults aged ≥ 60 years. Following a trend roughly opposite to the incidence of the event, in general, more children aged 0–9 years died from scorpionism in Brazil, with males in this age group being the highest, with 1.29 deaths (n = 247) per 1 million inhabitants. Lethality was similar to mortality rates, and both sexes had the highest mortality rates in children aged 0–9 years (36.7% of reported deaths).

Table 1. Number of cases and deaths of scorpionism and their respective incidence rates (per 100,000 inhabitant-years), mortality rates (per 1,000,000 inhabitant-years), and lethality (%) according to sex and age group in Brazil between 2012 and 2024.

| Sex | Rates | Age group | | | | | |
| --- | --- | --- | --- | --- | --- | --- | --- |
| | | 0–9 years | 10–19 years | 20–39 years | 40–59 years | ≥ 60 years | Total |
| Male | Incidence | 49,8 | 57,4 | 63,6 | 75,0 | 84,2 | 65,9 |
| | cases | 95,056 | 118,759 | 271,739 | 237,612 | 135,803 | 858,969 |
| | Mortality | 1.29 | 0.40 | 0.32 | 0.41 | 0.48 | 0.52 |
| | Lethality (%) | 0.26 | 0.07 | 0.05 | 0.05 | 0.06 | 0.08 |
| | deaths | 247 | 83 | 138 | 130 | 78 | 676 |
| Female | Incidence rate | 46,6 | 60,6 | 61,3 | 71,1 | 74,6 | 63,7 |
| | cases | 85,271 | 120,814 | 267,366 | 244,980 | 151,623 | 870,054 |
| | Mortality | 0.76 | 0.30 | 0.25 | 0.31 | 0.37 | 0.38 |
| | Lethality (%) | 0.24 | 0.05 | 0.04 | 0.04 | 0.05 | 0.06 |
| | deaths | 205 | 59 | 109 | 106 | 75 | 554 |
| Total | Incidence rate | 48,2 | 59.0 | 62,4 | 73,0 | 78,8 | 64,8 |
| | cases | 180,327 | 239,573 | 539,105 | 482,592 | 287,426 | 1,729,023 |
| | Mortality | 0.98 | 0.35 | 0.28 | 0.36 | 0.42 | 0.46 |
| | Lethality (%) | 0.25 | 0.06 | 0.05 | 0.05 | 0.05 | 0.07 |
| | deaths | 452 | 142 | 247 | 236 | 153 | 1230 |

**Table 2. Number of cases and deaths of scorpionism and their respective standardized incidence rates (per 100,000 inhabitant-years), mortality rates (per 1,000,000 inhabitant-years), and lethality (%) according to region and sex in Brazil between 2012 and 2024.**

| Sex | Std. rate | Regions | | | | | |
|---|---|---|---|---|---|---|---|
| | | North | Northeast | Midwest | Southeast | South | Total |
| Male | Incidence | 33,4 | 90,1 | 53,6 | 77,1 | 14,7 | 65,9 |
| | cases | 38,641 | 317,927 | 54,879 | 419,948 | 27,574 | 858,969 |
| | Mortality | 0.46 | 0.69 | 0.52 | 0.58 | 0.06 | 0.52 |
| | Lethality (%) | 0.14 | 0.08 | 0.10 | 0.07 | 0.04 | 0.08 |
| | deaths | 53 | 244 | 53 | 314 | 12 | 676 |
| Female | Incidence | 21,6 | 109,8 | 52,7 | 61,7 | 12,1 | 63,7 |
| | cases | 24,739 | 410,960 | 55,075 | 355,636 | 23,644 | 870,054 |
| | Mortality | 0.22 | 0.66 | 0.38 | 0.36 | 0.02 | 0.38 |
| | Lethality (%) | 0.11 | 0.06 | 0.08 | 0.06 | 0.02 | 0.06 |
| | deaths | 27 | 261 | 43 | 219 | 4 | 554 |
| Total | Incidence | 27,5 | 100,3 | 53,2 | 69,2 | 13,3 | 64,8 |
| | cases | 63,380 | 728,887 | 109,954 | 775,584 | 51,218 | 1,729,023 |
| | Mortality | 0.35 | 0.69 | 0.46 | 0.47 | 0.04 | 0.46 |
| | Lethality (%) | 0.13 | 0.07 | 0.09 | 0.07 | 0.03 | 0.07 |
| | deaths | 80 | 505 | 96 | 553 | 16 | 1230 |

Table 2 shows regional variations in scorpionism across Brazil, with the northeast and southeast accounting for 87% of all cases. The southern region had the lowest number of cases and deaths among all regions regardless of sex. The risk of scorpion sting was higher in males, except in the northeast, where the incidence rate was higher in females than in males.

Fig 2 presents the local empirical Bayesian rates of scorpionism in Brazilian municipalities. Rates ranged from 0 to 431 cases per 10,000 inhabitant-years, significantly increasing over time.

Local empirical Bayesian rates between 2012 and 2024 showed substantial growth in Southern Bahia and Northern Minas Gerais, with notable increases in Northwestern São Paulo and the state of Pará. Starting in 2018, visualization of rates became more accurate, making it clearer which areas were most significantly impacted by scorpionism. The southeast region of the country had the highest percentage of deaths (45%), followed by the northeastern (41%); The northeastern region had well-distributed deaths in its states, while the southeastern region concentrated cases in the states of Minas Gerais and São Paulo. The number of deaths per municipality in Brazil, represented in the last map of Fig 2, showed that fatal injuries occurred mainly in the southeast and northeastern regions, with a strong impact on the state of Minas Gerais and Bahia. However, it also showed municipalities with very high numbers over the 13-year period, such as Altamira/PA, which had 16 deaths, the highest of any municipality in Brazil.

The results of cluster analyses are presented in Figs 3 (purely spatial scan), 4 (spatiotemporal scan), 5 (spatial variation of the temporal trend), and 6 (purely temporal and seasonal scan). Fig 3 made it possible to observe the high- and low-risk regions for scorpionism. High-risk regions (represented by warm colors; RR > 1) were predominantly in the northeast and southeast areas. Minas Gerais, São Paulo and Bahia had the highest number of high-risk clusters, covering more than 50% of the state.

Low-risk regions (RR < 1) for scorpionism were present in almost the entire country, being most evident in the north, midwest, south, and north of the northeast regions. In general, low-risk areas exhibited a specific pattern in their locations, characterized by small clusters occurring at high frequencies. This pattern is particularly evident in all regions except in the northern and midwest region of Brazil. Even though the northeastern region (mainly around the states of Paraíba, Pernambuco and Alagoas) doesn't have the highest incidence rates, a purely spatial high-risk cluster has almost completely

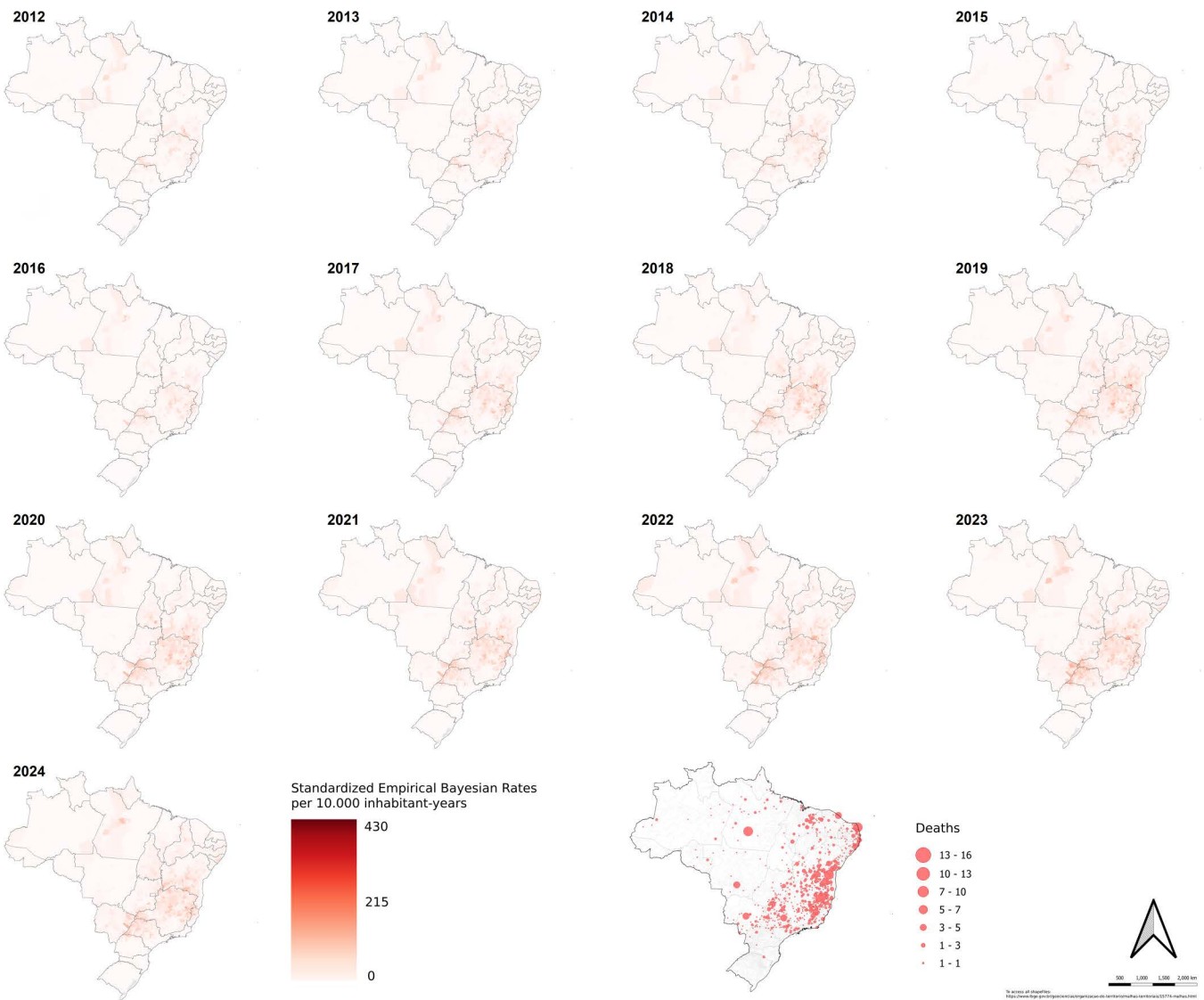

**Fig 2. Map of the local empirical Bayesian rates (10,000 inhabitants-year) and number of scorpionism deaths in Brazil between 2012 and 2024.** Source of base layer maps (states and municipality boundary): https://www.ibge.gov.br/geociencias/organizacao-do-territorio/malhas-territoriais/15774-malhas.html.

filled the region and raises concern because it is in an area with a high population concentration in the major coastal centers.

The low-risk spatio-temporal clusters (Fig 4) were not identified in this analysis. In contrast, high-risk spatiotemporal cluster began near the end of the study period, occurring between 2018 and 2024 in the northeastern and southeastern Brazilian regions, mainly in northern Bahia and Minas Gerais. The high-risk cluster (warm colors: RR > 1) were significant in the later periods (starting in 2018), representing the impact zones closer to the present time.

The spatial variation of the temporal trend cluster analysis between 2012 and 2024 (Fig 5) showed significant results for measuring the possibility of scorpion sting growth in specific regions represented by warm colors. The blue area, only in the northeast, showed negative trends, suggesting a possible reduction in cases. However, it is much smaller than the

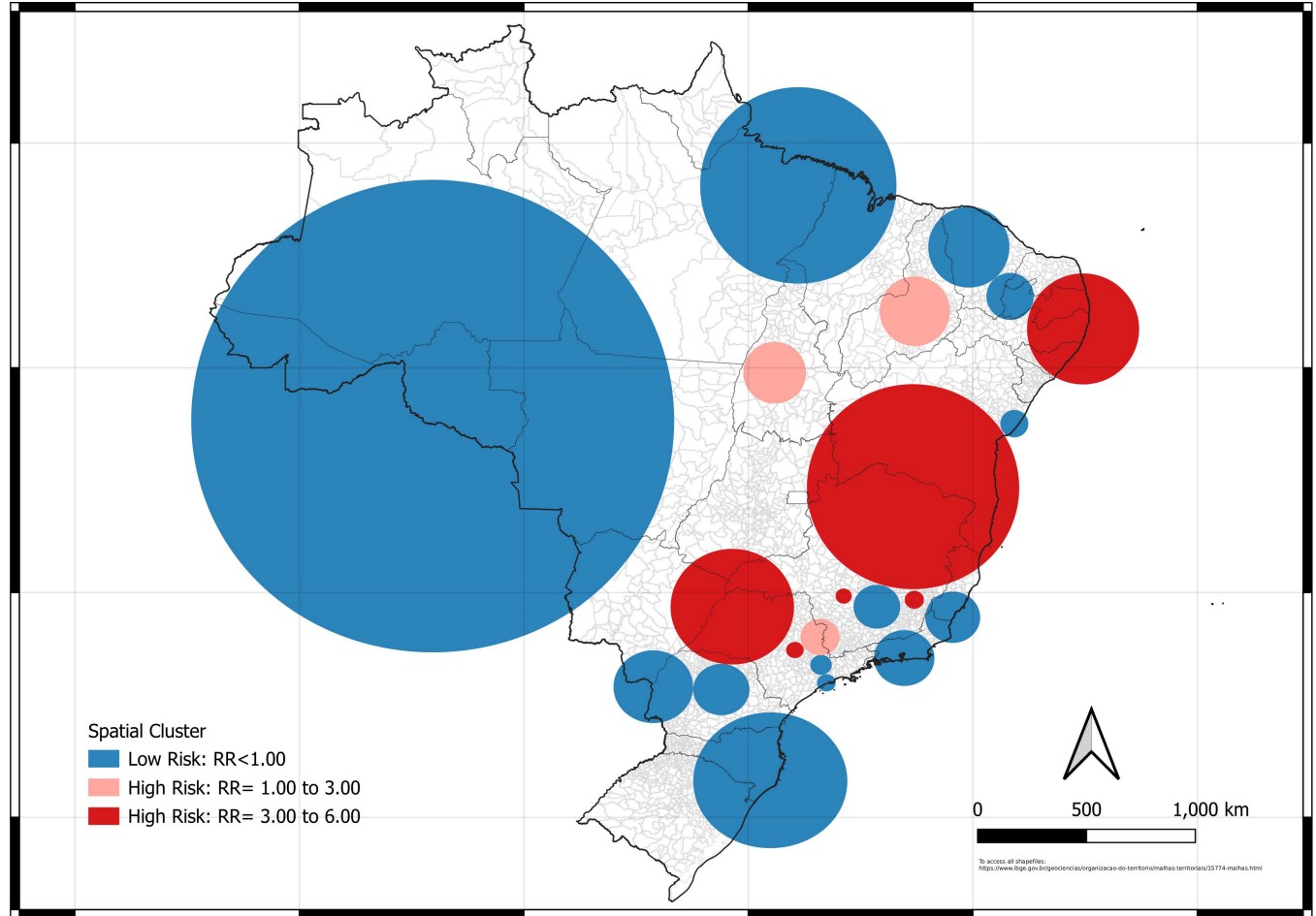

**Fig 3. Map of the purely spatial cluster of scorpionism in Brazil between 2012 and 2024.** Source of base layer maps (states and municipality boundary): https://www.ibge.gov.br/geociencias/organizacao-do-territorio/malhas-territoriais/15774-malhas.html.

other clusters in Brazil. The groups with a temporal trend of growth within them were particularly represented in very specific locations, representing all of Brazil's regions, with special attention to the north, which only had the state of Tocantins with a cluster showing an important trend. The regions with the most documented temporal growth trends were places in the southern, southeastern, and northeastern regions, with a projected yearly increase of 3–46%.

Independent of the temporal trend of the spatial clusters being positive or negative, the temporal trends outside them presented yearly values of approximately 8–10%, indicating a persistent increase in temporal trends in Brazil during the study period. Independent of the signal of the spatial-cluster temporal trends, the RR was greater or lower than 1. Regions with positive temporal trends, such as in the southeastern and northeastern regions, remain areas of high concern. In both clusters with trends > 12% (orange and red clusters), it is evident that the growth within the clusters is greater than those outside, which may influence the pattern of low-risk clusters, further raising the potential for future scorpion sting cases.

Fig 6 shows the purely temporal scan according to the months between January 2012 and December 2024 (A) and the seasonal scan between January and December (B). For the purely temporal cluster, it was possible to observe the cluster that began in July 2018 and continued until the end of the study period, representing significant growth in recent years and greater epidemiological concern. The seasonal data showed that the months of September to

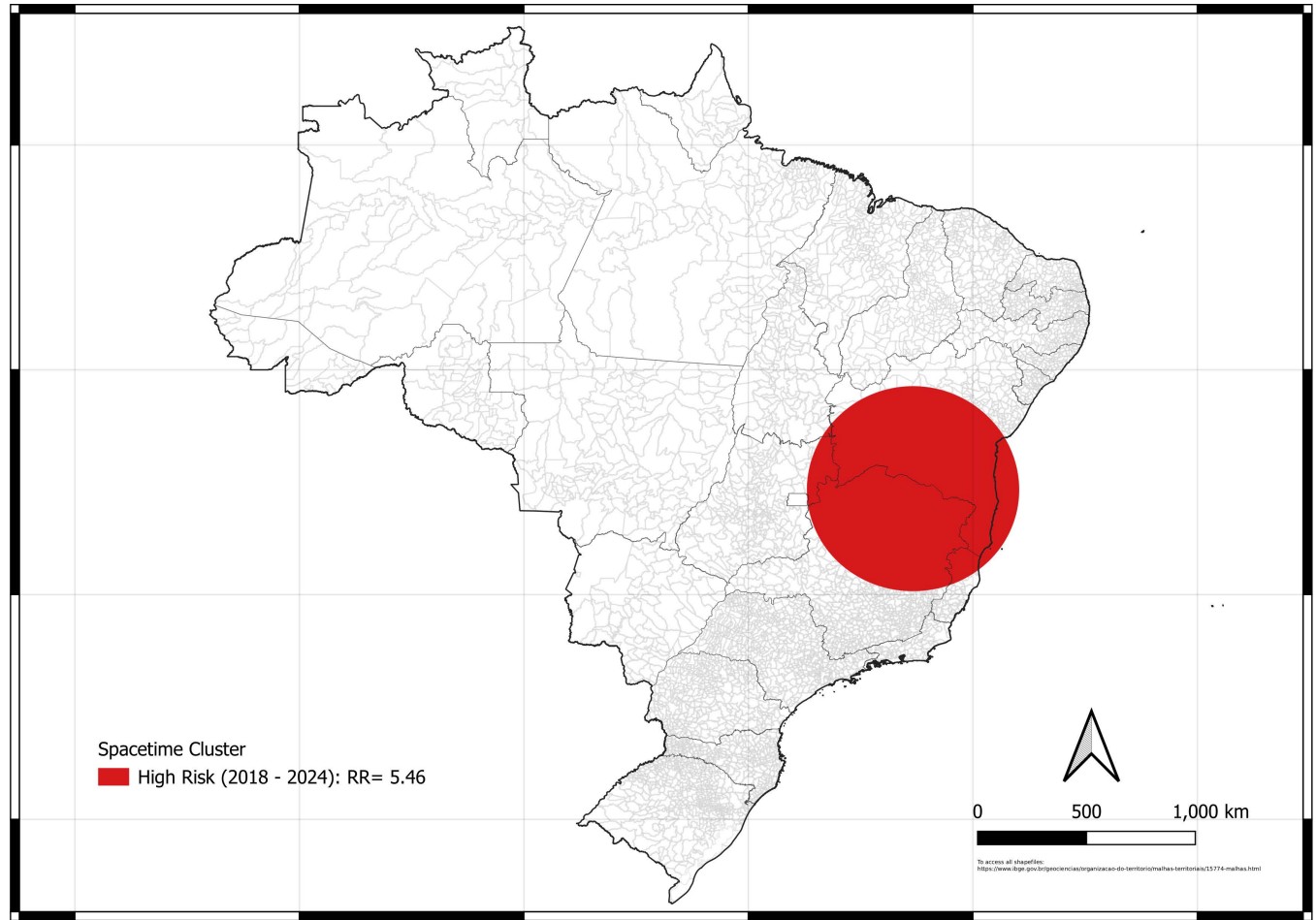

**Fig 4. Map of the space-time cluster of scorpionism in Brazil between 2012 and 2024.** Source of base layer maps (states and municipality boundary): https://www.ibge.gov.br/geociencias/organizacao-do-territorio/malhas-territoriais/15774-malhas.html.

December posed the highest risk for scorpionism in the total study period, particularly during the spring season. In contrast, colder periods, especially from June onwards, showed a lower occurrence of these events over the total study period.

Table 3 presents the observations of conditions within purely spatial clusters for comparison.

Among the observed variables, only urbanization and garbage collection showed non-significant results in the Student's *t-test*. Among the environmental variables, afforestation (natural vegetation) and NDVI had lower values in the high-risk spatial. In other words, where there is more vegetation there is less risk of accidents. Among the climatic variables, the maximum and minimum temperatures were higher in the high-risk clusters with lower rainfall/precipitation. The temperature range was also greater in places with a higher risk of accidents. Among the sociodemographic variables, however, it seems that places with better water and sewage treatment had a higher risk of accidents, which may be associated with the place where the animal stays. On the other hand, the literacy (alphabetization) factor - an indicator indirectly associated with income and quality of life - showed better values in low-risk locations, indicating an important sociodemographic relationship for scorpion accidents. Generally, the high-risk spatial clusters for scorpion accidents were characterized by warmer temperatures, less rainfall, less vegetation, and less alphabetization.

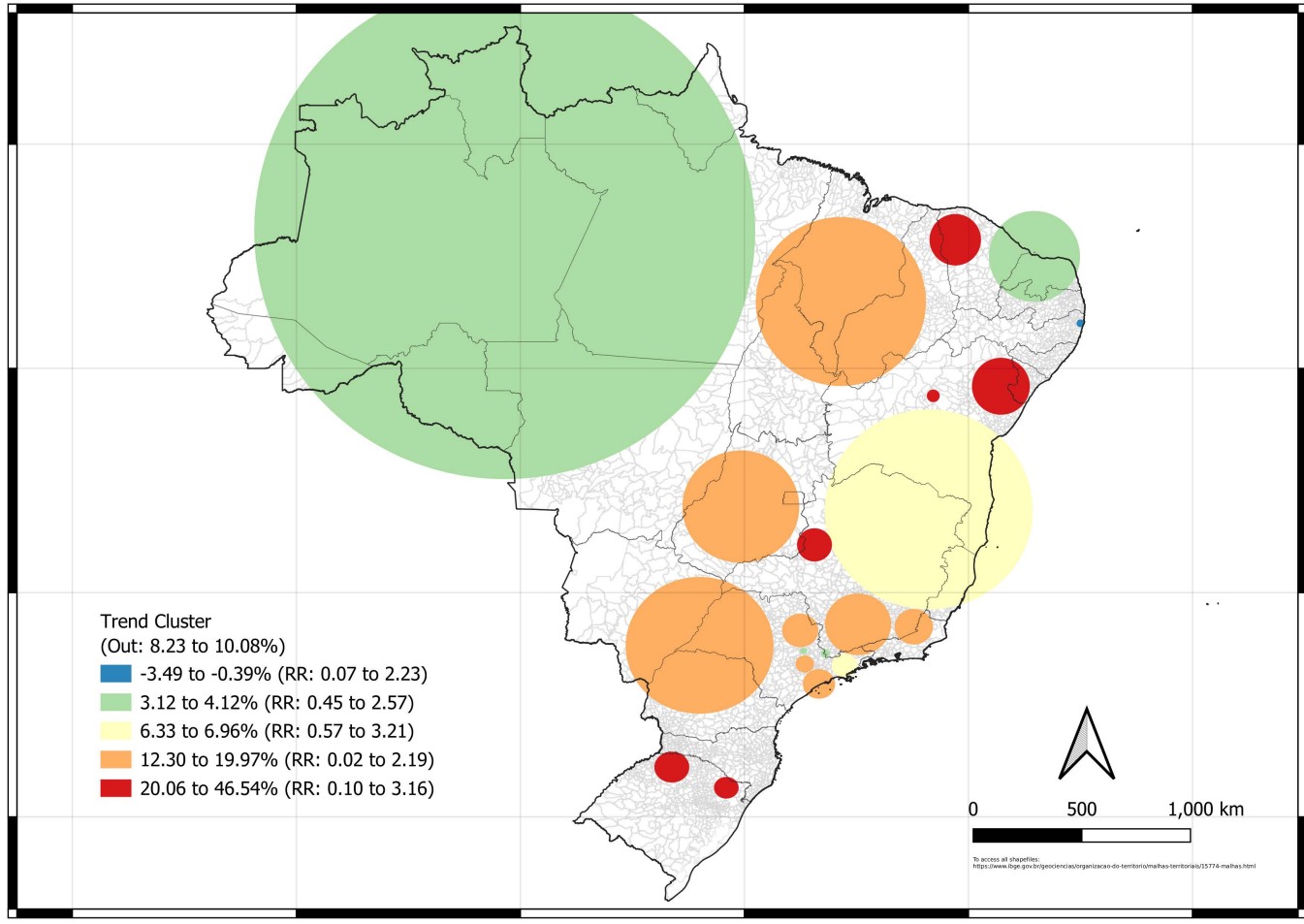

**Fig 5. Map of the spatial variation of temporal trend cluster of scorpionism in Brazil between 2012 and 2024.** Source of base layer maps (states and municipality boundary): https://www.ibge.gov.br/geociencias/organizacao-do-territorio/malhas-territoriais/15774-malhas.html.

## Discussion

Our data shows a significant increase in cases of scorpionism in Brazil, especially in the southeast and northeast, with an increase of 349% in 13 years of study. High mortality and lethality in children - especially in the north, high incidence in the elderly, higher incidence in women in the northeast and high temperature, low rainfall, vegetation and literacy were the main findings of this study.

Scorpionism is a growing public health problem in Brazil that primarily affects children, older adults, and the male population. As described in other studies, the economically active age group (20–59 years) and older adults (≥ 60 years) were most affected. Previous studies have suggested that the higher number of accidents in economically active age groups may be related to work or environmental factors [24,25]. In this age group, most poisonings were mild and required symptomatic treatment without specific serum therapies. In this study, as reported in previous studies, lethality was much higher in children [26,27], demonstrating the existence of a risk group requiring more attention and care. Timely antivenom treatment for these patients is crucial, as the longer the time between the bite and the first treatment, the higher the probability of death [28,29].

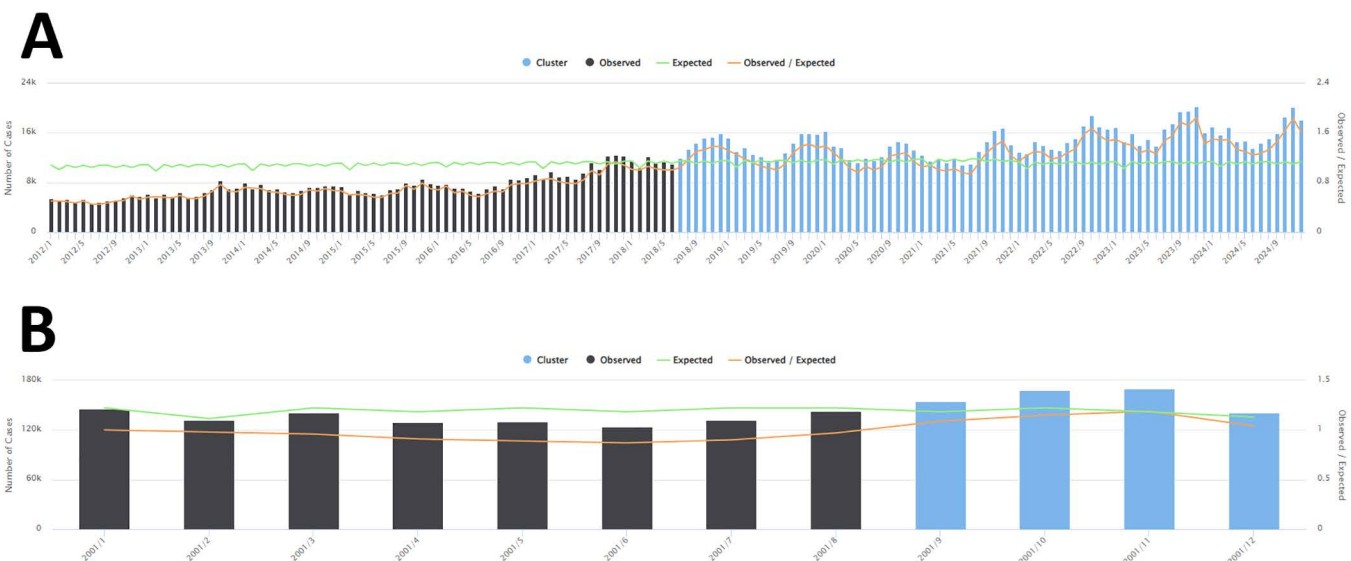

**Fig 6. Purely temporal scan for months (A) and seasonal scan (B) of scorpionism in Brazil between January 2012 and December 2024.**

**Table 3. Student's *t*-test of sociodemographic, environmental and climatic variables of high- and low-risk spatial clusters in Brazil.**

| Variables | High risk | Low risk | *p-value* |
|---|---|---|---|
| NDVI (%) | 0.54 | 0.61 | <0.000 |
| Minimum temperature (%) | 17.37 | 16.93 | <0.000 |
| Maximum temperature (°C) | 28.92 | 27.90 | <0.000 |
| Average precipitation (mm) | 82.64 | 117.15 | <0.000 |
| Temperature range | 11.54 | 10.97 | <0.000 |
| Sewage treatment (%) | 45.50 | 28.11 | <0.000 |
| Water treatment (%) | 72.29 | 67.89 | <0.000 |
| Garbage collection | 69.53 | 68.21 | 0.06 |
| Alphabetization | 85.47 | 89.72 | <0.000 |
| Urbanization (%) | 2.35 | 2.37 | 0.94 |
| Natural vegetation (%) | 13.41 | 27.02 | <0.000 |

Although the increase in incidence is related to an increase in age, younger children are more susceptible to severe envenomation and higher lethality. In the youngest age group, access to antivenom is fundamental for the prognosis of envenoming. Other studies have also shown that the younger the patient, the more severe the symptoms. Factors related to envenoming severity in children are attributed to the smaller body mass and size of the child and higher concentration levels of venom in blood plasma than in adults [28,30,31], and the immature immune response of children, which may influence the higher lethality in this age group [32].

Scorpionism is an essential public health concern across Brazil and is significant in almost all regions, particularly in the states of São Paulo, Minas Gerais, and Bahia, which have the most deaths. The northeastern region of Brazil has historically been affected by the presence of *T. stigmurus* species, and the southeastern region by *T. serrulatus*, both being responsible for most accidents in Brazil [33; 4].

Recent studies highlight the rise of scorpionism in Brazil as a growing health problem in recent decades [4,34], with essential reports in the northeast region in the states of Ceará [35], Paraíba [36], Rio Grande do Norte [37], and Bahia

[38,39]. In particular, the states of Bahia and Minas Gerais have played a leading role, with high incidence and death rates corroborated by the presence of high-risk clusters in recent years and significant annual growth, necessitating the need for effective preventive measures. The northeastern region around Pernambuco, Alagoas and Rio Grande do Norte raises concerns about the impact of accidents, located close to large urban centers and coastlines. [40] also identified a higher risk for women in the region - specifically in the state of Alagoas - accounting for an average of 59% of all recorded cases. The state of Alagoas is among the most incident in the country, reaching more than 270 cases per 100,000 inhabitants in this study and being evidenced by other studies [41], intensified during periods of agricultural activities.

Concerns about preventing scorpion sting envenomation in these regions are marked by the number of deaths among vulnerable populations, usually children up to the age of 9 years. Presently, the press has reported an increasing number of fatal cases of scorpionism, which has generated hypotheses about its occurrence in hot and hybrid weather areas and those with an accumulation of organic waste that attracts scorpions and their prey, such as cockroaches, larvae, and other insects [14].

The distribution of scorpions of medical interest in all regions of Brazil has been published in the national synanthropic control manuals [5]. [42] explained the adaptation and survival of these animals in different situations and environments, which may explain the survival characteristics of scorpions in urban areas. The findings showed that water deprivation is the main factor detrimental to survival and reproduction, which generally does not occur in urban areas, as they often reside in locations with abundant water and food sources, such as underground galleries and culverts. However, these data are crucial to understanding essential aspects of scorpions and developing cost-effective strategies.

The presence of predators such as domestic fowl, guinea fowl, and opossums can be seen as an initial control measure in places that are difficult to manage ecologically. [43] evaluated the primary characteristics of predation by domestic chickens. This shows that birds have the potential for biological control of arthropods without fatal events, which may favor local and specific actions where traditional management strategies are insufficient.

There are also issues with the use of chickens that need to be considered, such as the fact that chicken feces can serve as breeding grounds for phlebotomine larvae, increasing the risk of diseases like leishmaniasis. Another issue is their diurnal habits, while scorpions are nocturnal, limiting their effectiveness in scorpion control. Though chickens prey on scorpions, their use in scorpion control is not entirely effective and can cause other health problems [44,45]. Other animals, such as the Cururu toad (*Rhinella icterica*), have shown greater promise. The Cururu toad has been scientifically proven to prey on scorpions and is highly resistant to their venom [46]. This demonstrates that synanthropic animals can be considered suitable for scorpion control. To be regarded as a good predator of a particular prey, the predator must, among other characteristics, know how to defend itself against the prey and resist its venom. However, changes in habitats caused by humans (disorganized urban growth) have made it difficult for these animals to thrive in urban environments, and their domestic breeding is often controlled by legislation, disqualifying these animals as reliable biological control agents in cities.

As synanthropic animals, scorpions have adapted to the human environment and habitat and easily enter homes through sewers or places with little maintenance. Climatic and environmental factors also favor the presence and proliferation of this animal in these areas, as observed in previous studies [14,38,47]. In this study, urbanization did not differ between risk areas; however, its processes favored human contact with animals, mainly by breaking the fauna/flora chain and creating new unhealthy and low-vulnerability points. A hot, dry climate favors the proliferation of scorpions, which often avoid cold places. Easy access to water and food in urban areas underlies the prevalence of these animals in sewage systems, densely populated urban centers, and land/areas without waste and forest maintenance [48]. Since the mid-1930s, with industry gaining more weight, there has been greater migration and population expansion to urban centers, significantly increasing the size and size of the population [49]. Even without significant findings in this study or in the literature [Duarte et al. 2024], this has opened up new niches for the survival and permanence of animals in urban environments - associated with their ecological plasticity - which facilitates contact with the constantly expanding population.

Historically, the state of Minas Gerais has had characteristics conducive to scorpions' survival, mainly environmental and climatic factors. A study conducted in the 1990s [50] showed that the frequency of cases increased over the years and occurred mainly at warm and hybrid weather locations between October and January, which can also be observed in the current data.

The northeast region is generally more affected by the impact of these accidents on its population. Unlike other regions, the northeast has a higher risk of accidents among females than males. Although the mortality rates are similar, more females die in this region, which raises questions about where accidents occur and their social circumstances. Accidents involving scorpions in the domestic environment have historically been linked to domestic services performed by most females in Brazil, as well as to manual labor and greater physical exposure of males, as described in a recent nationwide study [Almeida, 2022]. For historical and social reasons, females are more exposed to cleaning and subordination in outdoor work environments, which may favor contact with waste and areas conducive to scorpion survival.

However, some regions in Brazil did not show upward growth and could even be representative of protective areas. For example, the northern territory, specifically the Amazon region, requires a more complex approach.

The Northern region of Brazil is notably characterized by the Amazon Rainforest, the largest tropical forest globally, which harbors approximately 13% of the world's scorpion species [51]. Difficulty accessing parts of the territory and underreporting of cases create significant barriers to understanding the true extent of scorpion accidents. The northern region of Brazil, despite not reporting high scorpionism cases (despite having the municipality with the highest number of deaths in the study period), faces significant challenges in accurately capturing data, which may obscure the actual situation. [52] investigated scorpion fauna in the Amazon region and its phylogenetic diversity. This describes the importance of understanding the different species present in the region and the clinical manifestations of venom linked to efficient and high-quality treatment measures, especially in places where immediate therapies are difficult to find. The information barriers that persist in these areas can be characterized by identifying low-risk areas in the states of Pará, Amazonia, and Mato Grosso, although many cases will never be reported to the health services.

The lethality of scorpions in the northern region is an important and worrying finding. The northern region almost doubles the two main regions with the highest number of cases and deaths from scorpionism, and little is documented about this reality. An additional difficulty for this issue is the fact that treatment and the existence of variations in toxins between the different species of medical importance [53], generating bottlenecks and problems regarding the effectiveness of treatment. In Brazil, the serum is produced from the venom of *T. serrulatus*, which is not as effective in neutralizing the cerebral muscle manifestations caused by some species in the northern region of Brazil, especially *T. obscurus* [54].

Despite the low number of cases and classification of the northern region of Brazil as a low-risk area for scorpionism, due to various gaps in access, assistance, and available professionals in this location, this assessment may not reflect the true extent of the problem. A notable example is the municipality of Altamira/PA, situated along the Xingu River, which reported 16 fatal cases, revealing the potential severity in specific areas. This region has already been marked by a fatal accident involving scorpions in children who had numerous difficulties getting around and accessing any government unit, where they resorted to the use of alternative medicine in an attempt to survive. At the time, it took 2 days by boat to get from the accident site to a primary health unit and an average of 5 days in the dry season, making any favorable prognosis impossible, especially in cases of severe poisoning [55].

Faced with these difficulties, spatial analysis is essential for identifying risk locations and assessing the local characteristics that impact scorpion accidents [14]. This knowledge provides information that allows for epidemiological surveillance and scorpion control efforts in the highest-risk locations, providing an opportunity for early intervention and prevention [13,23].

Epidemiological surveillance employing spatial analysis, specifically spatial scanning techniques, enhances the measurement of risks and identifies places of greater priority for the population's health [21]. Techniques with spatial and temporal methodologies allow areas of active focus that have moved over time to be analyzed, as well as those that show

increased risk in the future, providing information that can shape preventive health policies [56]. The application of local empirical Bayesian rates, which control random fluctuations, mitigating possible underreported municipalities, and space-time modeling strengthen the integration of social and epidemiological determinants and epidemiological surveillance in combating events that harm public health and, in the case of scorpionism, prevents this number from growing even more rampantly.

Epidemiological surveillance of scorpion accidents is a complex process that involves environmental, socioeconomic, and health variables, and actions are needed for efficient environmental management (hygiene and sanitation), combined with scorpion control through mechanical capture (the only effective method to date). Essential measures include home protection that blocks animals from entering properties and promoting information and educational actions for the population on scorpionism [57]. In addition to the actions mentioned above, it is essential to have an efficient healthcare network with treatment available in strategically distributed Reference Units in the municipalities so that victims can attend and receive serum therapy, when necessary, in the shortest possible time [27]. The distribution of serum in a networked system that serves points that are difficult to access and invests in training professionals to identify, manage, and treat accidents involving venomous animals is fundamental to reducing deaths, sequelae, disability pensions, and overload on the health and social security system [58]. The deaths recorded in Altamira/PA may indicate data that does not represent the reality of the location, since they could be overestimated [59]. In many cases, especially in the northern region and in areas with multiple access barriers, the characterization of accidents/deaths caused by scorpions is subjective, and the failure to report the event requires several quality analyses [18].

An example of an adaptation to the intense increase in the number of accidents and deaths caused by scorpions occurred in the state of São Paulo, where, as of 2019, there has been an operational restructuring for the timely care of scorpion victims. This restructuring redefined or implemented new Reference Units for antivenom treatment, in which antivenom vials for scorpion sting envenoming are available in municipalities to provide scorpion antivenom to those injured by scorpions, mainly children up to 10 years old, providing access to anyone within the state of São Paulo within 1.5 hours of receiving specific treatment [Guerra et al., 2021].

The data used is subject to underreporting in health information systems, especially in the compulsory scorpion sting notification forms of SINAN. Collecting real-world information in regions where access to the internet and land transportation are precarious creates barriers to accessing data that persist to this day. Using reliable information that reflects reality, it would be possible to estimate the high- and low-risk areas representative of each population and region, enabling health policies and funding to be targeted at strategic points with a real impact on this event.

Another limitation of this study is the lack of information on the species that causes scorpion poisoning. These data are crucial for differentiating the severity of cases and introducing species to new locations, which can have a negative impact as they are integrated with urban areas and come in contact with humans. Identifying the types of scorpions that cause accidents in different regions of Brazil can generate information for increasingly effective therapeutic approaches, such as identifying patterns of scorpion stings and identifying the types of scorpions that cause accidents in different regions of Brazil.

This study is fundamental to improving scientific knowledge about this neglected public health problem, providing an overview of scorpionism in Brazil and identifying high-risk areas. The geographical localization of priority areas favors the development of community activities, prevention, control, and monitoring, as well as the distribution of medicines and financial resources. They are essential for avoiding contact with scorpions and adequately attending to people who suffer accidents. These characteristics are fundamental in helping to reduce the growing incidence, mortality, and lethality (especially in children) of scorpion accidents, fund subsidies to maintain vulnerable regions and identify potential risk factors that increase the risk of serious accidents. Understanding how scorpionism differs in Brazil is essential for interregional measures to control the animal as well as actions at the municipal level since a large number of cases in some Brazilian states may be located in specific regions and in a heterogeneous manner, requiring unique measures for efficient control

and prevention procedures. The profile of deaths makes these actions more specific, as they favor the identification of the population most affected by severe cases of animal poisoning and those most in need of assistance in the shortest possible time, boosting the prevention of fatal cases, which are often avoided through serotherapy relocation strategies, health planning, and disease prevention.

## Supporting information

**S1 Data. Dataset scorpionism.**
(CSV)

## Acknowledgments

Special thanks to all the members of the venomous animal study group for their contributions to the study.

## Author contributions

**Conceptualization:** Alec Brian Lacerda, Denise Cândido, Fan Hui Wen, Gisele Dias Freitas, Flávio Santos Dourado, Francisco Chiaravalloti Neto.

**Data curation:** Flávio Santos Dourado.

**Formal analysis:** Alec Brian Lacerda.

**Funding acquisition:** Francisco Chiaravalloti Neto.

**Investigation:** Alec Brian Lacerda, Thiago Salomão de Azevedo, Flávio Santos Dourado, Francisco Chiaravalloti Neto.

**Methodology:** Alec Brian Lacerda, Francisco Chiaravalloti Neto.

**Project administration:** Francisco Chiaravalloti Neto.

**Supervision:** Francisco Chiaravalloti Neto.

**Validation:** Alec Brian Lacerda, Denise Cândido, Thiago Salomão de Azevedo, Fan Hui Wen, Gisele Dias Freitas, Flávio Santos Dourado, Francisco Chiaravalloti Neto.

**Visualization:** Alec Brian Lacerda, Denise Cândido, Thiago Salomão de Azevedo, Fan Hui Wen, Gisele Dias Freitas, Flávio Santos Dourado, Francisco Chiaravalloti Neto.

**Writing – original draft:** Alec Brian Lacerda, Denise Cândido, Thiago Salomão de Azevedo, Fan Hui Wen, Gisele Dias Freitas, Flávio Santos Dourado, Francisco Chiaravalloti Neto.

**Writing – review & editing:** Alec Brian Lacerda, Denise Cândido, Thiago Salomão de Azevedo, Fan Hui Wen, Gisele Dias Freitas, Flávio Santos Dourado, Francisco Chiaravalloti Neto.

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
