## [Decision Letter · Decision Letter 0]

25 Feb 2025

Scorpionism in Brazil: Space-time approach and risk areas in 2012 to 2021

Dear Dr. Lacerda,

Thank you for submitting your manuscript to PLOS Neglected Tropical Diseases. After careful consideration, we feel that it has merit but does not fully meet PLOS Neglected Tropical Diseases's publication criteria as it currently stands. Therefore, we invite you to submit a revised version of the manuscript that addresses the points raised during the review process.

Please submit your revised manuscript within 60 days Apr 26 2025 11:59PM. If you will need more time than this to complete your revisions, please reply to this message or contact the journal office at plosntds@plos.org. Please include the following items when submitting your revised manuscript:

We look forward to receiving your revised manuscript.

Kind regards,

Wuelton Monteiro, Ph.D.

Section Editor

Wuelton Monteiro

Section Editor

Shaden Kamhawi

co-Editor-in-Chief

Paul Brindley

co-Editor-in-Chief

**Journal Requirements:**

1) Please provide an Author Summary. This should appear in your manuscript between the Abstract (if applicable) and the Introduction, and should be 150-200 words long. The aim should be to make your findings accessible to a wide audience that includes both scientists and non-scientists. Sample summaries can be found on our website under Submission Guidelines:

2) Some material included in your submission may be copyrighted. According to PLOSu2019s copyright policy, authors who use figures or other material (e.g., graphics, clipart, maps) from another author or copyright holder must demonstrate or obtain permission to publish this material under the Creative Commons Attribution 4.0 International (CC BY 4.0) License used by PLOS journals. Please closely review the details of PLOSu2019s copyright requirements here: PLOS Licenses and Copyright. If you need to request permissions from a copyright holder, you may use PLOS's Copyright Content Permission form.

Potential Copyright Issues:

- Figures 1, 2, 3, 4, and 5. Please provide a direct link to the base layer of the map (i.e., the country or region border shape) and ensure this is also included in the figure legend; and provide a link to the terms of use / license information for the base layer image or shapefile. We cannot publish proprietary or copyrighted maps (e.g. Google Maps, Mapquest) and the terms of use for your map base layer must be compatible with our CC BY 4.0 license.

3) We note that your Data Availability Statement is currently as follows: "The data will be made available in aggregate form, without any personal or identifiable information, for the entire period and at all study sites by means of a spreadsheet in xlsx format.". Please confirm at this time whether or not your submission contains all raw data required to replicate the results of your study. Authors must share the “minimal data set” for their submission. PLOS defines the minimal data set to consist of the data required to replicate all study findings reported in the article, as well as related metadata and methods (https://journals.plos.org/plosone/s/data-availability#loc-minimal-data-set-definition).

- The points extracted from images for analysis..

4) Please ensure that the funders and grant numbers match between the Financial Disclosure field and the Funding Information tab in your submission form. Note that the funders must be provided in the same order in both places as well.

**Reviewers' Comments:**

Reviewer's Responses to Questions

**Key Review Criteria Required for Acceptance?**

**Methods**

-Are the objectives of the study clearly articulated with a clear testable hypothesis stated?

-Is the study design appropriate to address the stated objectives?

-Is the population clearly described and appropriate for the hypothesis being tested?

-Is the sample size sufficient to ensure adequate power to address the hypothesis being tested?

-Were correct statistical analysis used to support conclusions?

-Are there concerns about ethical or regulatory requirements being met?

Reviewer #1: The study appears to be well-executed; however, the Materials and Methods section requires additional details to ensure reproducibility. Additionally, some of the analyses are not suitable for the type of data used.

Reviewer #2: (No Response)

Reviewer #3: (No Response)

Reviewer #4: Methods clearly described and appropriated

**Results**

-Does the analysis presented match the analysis plan?

-Are the results clearly and completely presented?

-Are the figures (Tables, Images) of sufficient quality for clarity?

Reviewer #1: The presentation of the results should be improved, as some figures have poorly informative legends.

Reviewer #2: (No Response)

Reviewer #3: • In Tables 1 and 2, the differences between the two incidence indicators and between the variable "cases" and "n" are not clear. Additionally, the table layout would be clearer if the first column contained the indicators, followed by three more columns for data presentation: one for women, one for men, and another for overall data.

• In the paragraphs beginning on lines 140 and 148, Table 1 should be referenced.

• In Table 3, "T-test (mean values)" should be removed from the matrix row as it creates confusion regarding the data presented. It is also unnecessary to include the footnote "* not significant."

Reviewer #4: Results are clearly presented

**Conclusions**

-Are the conclusions supported by the data presented?

-Are the limitations of analysis clearly described?

-Do the authors discuss how these data can be helpful to advance our understanding of the topic under study?

-Is public health relevance addressed?

Reviewer #1: The conclusions are appropriate. However, certain points were not addressed, such as the high lethality in the North and the significant accident risk in the states of Sergipe/Alagoas.

Reviewer #2: (No Response)

Reviewer #3: • In the discussion, it would be important to elaborate on the differences in lethality between regions, as the North has approximately twice the porcentage for both men and women (Table 2), which could be related to North-South inequalities in Brazil (the North is more economically disadvantaged).

Reviewer #4: Conclusions supported by data

**Editorial and Data Presentation Modifications?**

Reviewer #1: (No Response)

Reviewer #2: (No Response)

Reviewer #3: • The bibliography is somewhat outdated; only 15 of the 48 references are from the last five years, and none are from the past year, despite existing scientific production in the field (40 references with "Scorpion Stings"[Mesh] in PubMed for the last year). The bibliographic references are not listed in alphabetical order by author.

• Although it is part of the study’s objectives, it is striking that the Introduction does not mention the relationship between the expansion of scorpionism and climate change, which should already be noted in this section. An interesting work to consult is:

Lira, A.F.d.A., Badillo-Montaño, R., Lira-Noriega, A., and de Albuquerque, C.M.R. (2020), Potential distribution patterns of scorpions in north-eastern Brazil under scenarios of future climate change. Austral Ecology, 45: 215-228. https://doi.org/10.1111/aec.12849

Reviewer #4: Minor revision

**Summary and General Comments**

Reviewer #1: The work titled "Scorpionism in Brazil: Space-time approach and risk areas in 2012 to 2021" addresses a highly interesting topic, which is scorpionism in Brazil. I really enjoyed reading this manuscript. However, I have some suggestions that I believe would be important for the authors to incorporate, and there is missing information in the Materials and Methods section that is necessary for the study to be reproducible.

The first major point is that I believe it would be beneficial for the authors to extend the study period and include the available data from 2022, 2023, and 2024. This would allow for an updated version that could serve as a reference for future studies.

The second major point concerns the statistical analyses using the t-test. Does your response variable follow a normal distribution? I ask this because I believe the variables used are unlikely to have a normal distribution, meaning that comparing means may not be the best strategy.

Others comments:

Materials and Methods:

• Lines 91-95 and 96-98: The authors should better explain what Local Empirical Bayesian is, what the criterion type I refers to, which version of R was used, and whether any specific packages were applied. Additionally, what is the retrospective Poisson model? Without this information, someone unfamiliar with the topic will find it difficult to replicate the study.

• Line 115: The authors should clarify where the variables were obtained from and at what resolution. There is a lack of information necessary for replication. Were the values obtained from the centroid of the municipality? Was an average used? What software was used to process the variables?

Results:

• Figure 6: A better explanation is needed. The legend does not even indicate what panel B represents.

Discussion:

• I believe there should be a discussion about the lethality in northern Brazil, as the data presented in Table 2 clearly show that it is almost twice as high as in other regions.

• Another point: Observing the map in Figure 3, Sergipe/Alagoas also appears as a high-risk region. However, throughout the text, this is not mentioned, and the focus is solely on Bahia. Why is that?

Reviewer #2: The paper provides a comprehensive overview of scorpionism in Brazil, emphasizing its public health significance, epidemiological trends, and associated risk factors. However, several aspects of the manuscript require improvement to enhance clarity and depth of analysis. Below are key recommendations for revision:

1- The text should undergo a thorough review to refine English language usage, ensuring precise scientific terminology. Grammatical errors should be corrected, such as the incorrect use of singular verbs with plural nouns (e.g., "Data" in line 83 should take a plural verb). The use of non-standard or uncommon word combinations should be revised to improve readability and maintain scientific accuracy.

2- While the introduction provides an adequate overview of scorpionism in Brazil, it would benefit from a broader discussion of its global impact. Incorporating a brief comparison with other endemic regions would help contextualize the significance of scorpionism in Brazil relative to worldwide trends.

3- The paper lacks sufficient detail on venom composition and clinical effects of different scorpion species. A brief discussion of interspecies differences in venom toxicity, their physiological impact, and clinical manifestations would strengthen the scientific foundation of the study.

4- The study does not clarify whether confounding factors such as socioeconomic status and healthcare access were controlled for in the analysis.

5- The abstract should be structured with subheadings (e.g., Background, Methods, Results, and Conclusion) to improve clarity and readability.

6- The funding acknowledgment should be moved to a separate section rather than being included in the methodology.

7- The discussion section should be reorganized for better logical flow. Redundant information should be removed or consolidated, such as the multiple mentions of antivenom treatment, which should be grouped into a dedicated section.

8- The role of biological control methods (e.g., chickens, Cururu toads) is an interesting aspect but currently lacks sufficient depth. Incorporating evidence-based discussions, such as documented studies on their efficacy, would enhance this section.

Reviewer #3: This is a very interesting study, with a highly rigorous methodology and great relevance to public health in the context of global climate change

Reviewer #4: Dear authors,

Congratulations for this beautiful work!

The only thing that I will suggest to you is to add a summary of your results in the discussion. It will help us to remember all results presented in the paper. Therefore, consider the it as the first paragraph of discussion.

Congrats once again!

Best regards,

PLOS authors have the option to publish the peer review history of their article (what does this mean? ). If published, this will include your full peer review and any attached files.

**Do you want your identity to be public for this peer review?** For information about this choice, including consent withdrawal, please see our Privacy Policy .

Reviewer #1: No

Reviewer #2: No

Reviewer #3: No

Reviewer #4: No

**Figure resubmission:**

**Reproducibility:**



---

## [Editor Report · Decision Letter 1]

25 Sep 2025

Dear Mr. Lacerda,

We are pleased to inform you that your manuscript 'Scorpionism in Brazil: Space-time approach and risk areas in 2012 to 2024' has been provisionally accepted for publication in PLOS Neglected Tropical Diseases.

Best regards,

Wuelton Monteiro, Ph.D.

Section Editor

Wuelton Monteiro

Section Editor

Shaden Kamhawi

co-Editor-in-Chief

Paul Brindley

co-Editor-in-Chief

---

## [Editor Report · Acceptance letter]

Dear Mr. Lacerda,

We are delighted to inform you that your manuscript, "Scorpionism in Brazil: Space-time approach and risk areas in 2012 to 2024," has been formally accepted for publication in PLOS Neglected Tropical Diseases.

Best regards,

Shaden Kamhawi

co-Editor-in-Chief

Paul Brindley

co-Editor-in-Chief
